# Exponential Stability of Fractional Large-Scale Neutral Stochastic Delay Systems with Fractional Brownian Motion

T. Sathiyaraj [1,*] , T. Ambika [2] and Ong Seng Huat [1,3]

1    Institute of Actuarial Science and Data Analytics, UCSI University, Kuala Lumpur 56000, Malaysia
2    Department of Computer Science, Rev. Jacob Memorial Christian College, Dindigul 624612, India
3    Institute of Mathematical Sciences, University of Malaya, Kuala Lumpur 50603, Malaysia
*    Correspondence: sathiyaraj133@gmail.com

**Abstract:** Mathematics plays an important role in many fields of finance. In particular, it presents theories and tools widely used in all areas of finance. Moreover, fractional Brownian motion (fBm) and related stochastic systems have been used to model stock prices and other phenomena in finance due to the long memory property of such systems. This manuscript provides the exponential stability of fractional-order Large-Scale neutral stochastic delay systems with fBm. Based on fractional calculus (FC), $\mathbb{R}^n$ stochastic space and Banach fixed point theory, sufficiently useful conditions are derived for the existence of solution and exponential stability results. In this study, we tackle the nonlinear terms of the considered systems by applying local assumptions. Finally, to verify the theoretical results, a numerical simulation is provided.

**Keywords:** dynamic risk in asset pricing; exponential stability; finance modeling and derivatives; fractional calculus; fractional Brownian motion; large dimensional problems; simulation and computation in long short-term memory; time delay

## 1. Introduction

Knowledge of mathematics, probability, statistics, and other analytical approaches is essential to develop methods and theories in finance and to test their validity through analysis of empirical real-world data. For example, mathematics, probability, and statistics help develop pricing models for financial assets such as stocks, bonds, currencies, and derivative securities and propose financially optimal strategies to decision makers based on their preferences. Brownian motion is a mathematical process used to describe random fluctuations in the stock market. It assumes that stock prices move randomly and follow a random walk. It is a type of stochastic process which can often be seen to model the movement of particles in a fluid or gas. However, Brownian motion is widely used in finance to model the random walk of stock prices over time. To apply Brownian motion in stock market modeling, the randomness of the price movement is used, as there is no particular trend and direction. This randomness is then modeled as a series of random steps, where each step represents a small change in the stock price. The size of each step is determined by the stock volatility, which is a measure of how much the stock price tends to oscillate over time. One important feature of Brownian motion is that it is a continuous process, meaning that the stock price can take on any value within a certain range. This makes it useful for modeling the behavior of stock prices over time, as it allows us to capture the full range of possible outcomes. However, while Brownian motion can be a useful tool for understanding the behavior of stock prices, it is not a perfect model. Stock prices can be influenced by a wide range of factors, including news events, company performance, and economic conditions. These factors can cause stock prices to move in ways that are not easily captured by a simple model such as Brownian motion.

The Hurst index has recently been introduced as a useful tool for assessing the memory effect, frequently measured by the autocorrelation function Hurst (1951). $\mathcal{H}(0 < \mathcal{H} < 1)$ is a common way to represent the Hurst index.

(1) When $0 < \mathcal{H} < 0.5$, the time series exhibits a negative correlation and antipersistent behaviour, or short-dependence memory.
(2) When $\mathcal{H} = 0.5$, the time series is independent.
(3) When $0.5 < \mathcal{H} < 1$, the time series exhibits persistent behaviour, or long-dependence memory.

The concept of fractional derivatives is not new, and FC has a long history of up to three centuries. The number of FC-related publications increased significantly in the later decades and mid-20th century. One of explanations for the high level of curiosity in fractional differential equations (FDEs) is that they can be used to define a diverse range of physical Hilfer (2000), chemical Oldham (2010), and biological Magin (2010) processes. Fractional derivative plays an important role in memory and hereditary processes. Several studies have been conducted to examine the long memory in the financial markets, since memory effect is a significant feature in financial systems. FC can be found in a variety of applications as a new branch of applied mathematics. Leibnitz, Caputo, Liouville, Riemann, Euler, and others are credited with a significant amount of foundational mathematical theory relevant to FC analysis. Nonetheless, throughout the last few decades, increasingly compelling representations have been discovered in numerous engineering and science disciplines (see Ortigueira (2011)). It should be highlighted that the existence hypothesis of FDEs is committed to a considerable part of the recent studies (see Balachandran et al. (2012); Nieto and Samet (2017); Singh et al. (2017); Tian and Nieto (2017)).

Recently, Bhaskar and Biswajit (2023) examined the effects of the steep surge in crude oil price shock on the stock price returns and currency exchange rates of G7 countries, namely Canada, France, Germany, Italy, Japan, the United Kingdom and the United States, in the context of the Russia–Ukraine conflict. Regime switches in the empirical relation between return dynamics and implied volatility in energy markets have been discussed in Okawa (2023). Optimal combination of proportional and Stop-Loss reinsurance with dependent claim and stochastic insurance premium have been studied in Sari et al. (2023). Herding trend in working capital management practices: evidence from the non-financial sector of Pakistan is analyzed in Farooq et al. (2023). Growth of venture firms under state capitalism with Chinese characteristics: qualitative comparative analysis of fuzzy set is discussed in Yun et al. (2023). In Li et al. (2014), the authors established a fractional-order stochastic differential equation model to describe the effect of trend memory in financial pricing.

While analyzing, there must be considerations for functional structures, ambient noise, and temporal delays, which can be quite valuable when constructing further sensible scientific models Mao (1997). The solution process for a stochastic fractional partial differential equation driven by space–time white noise has been studied in Wu (2011). The controllability of fractional and Hilfer fractional dynamical systems has been studied in Kumar et al. (2022a, 2022b, 2023). The relations between a singular system of differential equations and a system with delays, and stability of fractional-order quasi-linear impulsive integro-differential systems with multiple delays have been studied in Dassios (2022); Kalidass et al. (2022).

Another type of noise exposure is continuous. This can be modeled using Levy methods. In particular, methods based on Poisson random measures, as a common non-Gaussian stochastic method, have already received a lot of attention in a variety of fields and have been used to predict when demand for supply chain systems will increase Song (2009). Mathematical modeling of one-sever m-form random queuing in a network system is modeled in the stochastic environment problems Seo and Lee (2011), distribution patterns of phone users in the service area of wireless links Taheri et al. (2010), as well as other naturally occurring anomalies in a variety of areas Applebaum (2009). In Rockner and Zhang (2007) the existence, uniqueness and huge deviation principle solutions to jump

type stochastic evolution equations were investigated. Many researchers have recently turned to FDEs as a useful tool for describing a variety of steady physical processes.

However, research into nonlinear FDE stability theory is still in its early phases, and much more work in this field is possible. Recently, the theoretical notion of FDEs was thoroughly investigated, yielding several fundamental discoveries, including the stability theory. In mathematical terms, stability theory is concerned with the convergence of differential equation solutions under minor changes in the original data. The topic of stability is critical in the study of FDEs, and many writers have addressed it (see Ahmed et al. (2007); Gao and Yu (2005); Odibat (2010); Wang et al. (2012)). In any event, nonlinear FDEs are more difficult to analyze for stability than conventional integer-order differential equations. Many authors have been drawn to the study of nonlinear FDE stability theory during the last few decades, and as a result, numerous approaches have been created. However, it is important to emphasize that just a few steps have been carried out to study the durability of FDEs using fixed point theorems. Burton and Zhang (2012) began a thorough investigation of the stability properties of differential equations using fixed point theorems. Following that, several authors used the fixed point method to establish sufficient conditions for the stability of the differential systems (see Ren et al. (2017); Shen et al. (2020)). Based on the above discussions, the exponential stability of FDEs with order $\tilde{\alpha} \in (\frac{1}{2}, 1)$ is considered through a fixed point approach. It is envisaged that FDEs with fBM will be important for modeling the chaotic behavior of stock prices and financial instruments. The exponential stability of FDEs is an important property in analysis and application in financial systems.

This paper's main contributions are as follows:

(i) A nonlinear fractional Large-Scale neutral stochastic delay system (NFSDS) is considered in $\mathbb{R}^n$ stochastic settings.
(ii) To determine the existence and uniqueness of a solution, the fixed point theorem and local assumptions on the nonlinear portion are utilized.
(iii) The stability and exponential stability of a certain NFSDS are established by the use of Hölder inequality and Gronwall's inequality.

The following assertions outline the paper's innovations and challenges and future direction:

(i) Stability and exponential stability results for NFSDS are new in $\mathbb{R}^n$ stochastic settings.
(ii) Study of the exponential stability of the proposed system is not easy, taking the norm estimation on nonlinear stochastic and Large-Scale neutral as the terms used in this paper.
(iii) It is more difficult to validate the system's weaker assumptions (1).

The following is an outline of the study: In Section 2, the model description and prelims are given. Our major findings are proved in Sections 3 and 4. Finally, Section 5 presents an illustration of the theory and Section 6 draws a conclusion.

## 2. System Description and Preliminaries

Consider the following NFSDS given by

$$
\begin{aligned}
{}^{\mathcal{C}}\tilde{\bar{\mathcal{D}}}^{\tilde{\alpha}}\left[\mathsf{x}_\mathsf{l}(\mathsf{t}) - \tilde{\mathsf{g}}_\mathsf{l}(\mathsf{t}, \mathsf{x}_\mathsf{l}(\mathsf{t}), \mathsf{x}_\mathsf{l}(\mathsf{t} - \tilde{\mathsf{h}}(\mathsf{t})))\right] =& \tilde{\bar{\mathcal{A}}}_l \mathsf{x}_\mathsf{l}(\mathsf{t}) + \tilde{\mathsf{f}}_\mathsf{l}(\mathsf{t}, \mathsf{x}_\mathsf{l}(\mathsf{t}), \mathsf{x}_\mathsf{l}(\mathsf{t} - \tilde{\mathsf{h}}(\mathsf{t}))) \\
& + \int_0^{\mathsf{t}} \tilde{\sigma}_l(\mathsf{s}, \mathsf{x}_\mathsf{l}(\mathsf{s}), \mathsf{x}_\mathsf{l}(\mathsf{s} - \tilde{\mathsf{h}}(\mathsf{s}))) dw(\mathsf{s}) \\
& + \int_0^{\mathsf{t}} \tilde{\eta}_l(\mathsf{s}, \mathsf{x}_\mathsf{l}(\mathsf{s}), \mathsf{x}_\mathsf{l}(\mathsf{s} - \tilde{\mathsf{h}}(\mathsf{s}))) dw_{(\mathsf{s})}^{\mathcal{H}}, \\
\mathsf{x}_\mathsf{l}(\mathsf{t}) =& \varphi(\mathsf{t}), \quad \mathsf{t} \in [-h, 0],
\end{aligned}
\tag{1}
$$

where $\mathsf{t} \in [0, T]$, $\frac{1}{2} < \tilde{\alpha} < 1$, $\mathsf{x}_\mathsf{l}(\mathsf{t}) \in \mathbb{R}^{n_l}$ ($l = 1$ to $N$), $\ni \sum_{l=1}^{N} n_l = n$ and $\tilde{\bar{\mathcal{A}}}_l$ is $n_l \times n_l$ continuous matrix valued functions. Define $\mathcal{C}^{n_l} = \mathcal{C}([-h, 0], \mathbb{R}^{n_l})$, a Banach space of continuous functions mapping from $[-h, 0] \to \mathbb{R}^{n_l}$. Define $[0, T] := J$, Further, $\tilde{\mathsf{g}}_\mathsf{l} : J \times \mathcal{C}^{n_l} \times \mathcal{C}^{n_l} \longrightarrow \mathbb{R}^{n_l}$, $\tilde{\mathsf{f}}_\mathsf{l} : J \times \mathcal{C}^{n_l} \times \mathcal{C}^{n_l} \longrightarrow \mathbb{R}^{n_l}$, $\tilde{\sigma}_l : J \times \mathcal{C}^{n_l} \times \mathcal{C}^{n_l} \longrightarrow \mathbb{R}^{n_l \times n_l}$, $\tilde{\eta}_l : J \times \mathcal{C}^{n_l} \times \mathcal{C}^{n_l} \longrightarrow$

$\mathbb{R}^{n_l \times n_l}$ are continuous functions which will be specified in the future. Moreover, $w_{(s)}^{\mathcal{H}}$ is a fBm with $\mathcal{H} \in (\frac{1}{2}, 1)$ which is defined by its stochastic representation

$$w_{(s)}^{\mathcal{H}} := \frac{1}{\Gamma\left(\mathcal{H} + \frac{1}{2}\right)} \left( \int_{-\infty}^{0} [(t-s)^{\mathcal{H}-\frac{1}{2}} - (-s)^{\mathcal{H}-\frac{1}{2}}] dw(s) + \int_{0}^{t} (t-s)^{\mathcal{H}-\frac{1}{2}} dw(s) \right)$$

here $\Gamma$ denotes the Gamma function $\Gamma(\alpha) := \int_{0}^{\infty} y^{\alpha-1} \exp(-y) dy$ and $0 < \mathcal{H} < 1$ is called the Hurst parameter (one can see the connection with the Hurst parameter for self-similar processes).

Let us consider a probability space $(\Omega, \mathcal{F}, P)$ with a probability measure $P$ and $w(t) = (w_1(t), w_2(t), \ldots, w_n(t))^{\mathcal{T}}$ be an $n-$dimensional Wiener process defined on $(\Omega, \mathcal{F}, P)$. Let $\{\mathcal{F}_t / t \in J\}$ be the filtration generated by $\left\{ w(s), w_{(s)}^{\mathcal{H}} : 0 \leq s \leq t \right\}$ defined on $(\Omega, \mathcal{F}, P)$. Let $L_2(\Omega, \mathcal{F}_t, \mathbb{R}^{n_l})$ denote the Hilbert space of all $\mathcal{F}_t$-measurable square integrable random variables with values in $\mathbb{R}^{n_l}$. Let $L_2^{\mathcal{F}}(J, \mathbb{R}^{n_l})$ be the Hilbert space of all square integrable and $\mathcal{F}_t$-measurable processes with values of $\mathbb{R}^{n_l}$. Let $\mathcal{B} = \left\{ x_l(t) : x_l(t) \in C(J, L_2(\Omega, \mathcal{F}_t, \mathbb{R}^{n_l})) \right\}$ be a Banach space of all continuous square integrable and $\mathcal{F}_t$-adapted processes with norm $\|x_l\|^2 = \sup_{t \in J} \mathbb{E}\|x_l(t)\|^2$ and $\|\varphi\|^2 = \max\{\mathbb{E}\|\varphi(t)\|^2 : t \in [-h, 0]\}$ for any $t \geq 0$, any given $\varphi \in C([-h, 0], \mathbb{R}^{n_l})$ denotes the Banach space of continuous functions mapping from $[-h, 0]$ to $\mathbb{R}^{n_l}$. For more details on fractional calculus definitions, stochastic theory and fBm, one can read our published paper Balasubramaniam et al. (2020); Sathiyaraj and Balasubramaniam (2018); Sathiyaraj et al. (2019).

**Definition 1.** *The Riemann–Liouville fractional operators (left sided) for $\tilde{n} - 1 < \tilde{\alpha} < \tilde{n}$ for $f_l : [0, \infty) \to \mathbb{R}$ are as follows:*

$$(I_{0+}^{\tilde{\alpha}} f_l)(\tilde{x}_l) = \frac{1}{\Gamma(\tilde{\alpha})} \int_{0}^{\tilde{x}_l} (\tilde{x}_l - t)^{\tilde{\alpha}-1} f_l(t) dt.$$

$$(D_{0+}^{\tilde{\alpha}} f_l)(\tilde{x}_l) = D^{\tilde{n}} (I_{0+}^{\tilde{n}-\tilde{\alpha}} f_l)(\tilde{x}_l).$$

**Definition 2.** *Podlubny (1998): The Caputo derivative for $\tilde{n} - 1 < \tilde{\alpha} < \tilde{n}$ for $f_l : [0, \infty) \to \mathbb{R}$ is as follows:*

$$^{\mathcal{C}} D_t^{\tilde{\alpha}} f_l(t) = \frac{1}{\Gamma(\tilde{n} - \tilde{\alpha})} \int_{0}^{t} \frac{f_l^{\tilde{n}}(s)}{(t-s)^{\tilde{\alpha}-\tilde{n}+1}} ds.$$

*and its Laplace transform is*

$$\mathcal{L}\{^{\mathcal{C}} D_t^{\tilde{\alpha}} f_l(t)\}(s) = s^{\tilde{\alpha}} f_l(s) - \sum_{l=0}^{\tilde{n}-1} f^l(0^+) s^{\tilde{\alpha}-1-l}.$$

**Definition 3.** *Podlubny (1998): The two-parameter family of Mittag–Leffler function is given by*

$$\mathcal{E}_{\tilde{\alpha}, \beta}(z) = \sum_{l=0}^{\infty} \frac{z^l}{\Gamma(l\tilde{\alpha} + \beta)} \quad \text{for } \tilde{\alpha}, \beta > 0.$$

*The general Mittag–Leffler function satisfies the below identity*

$$\int_{0}^{\infty} e^{-t} t^{\beta-1} \mathcal{E}_{\tilde{\alpha}, \beta}(t^{\tilde{\alpha}} z) dt = \frac{1}{1-z} \quad \text{for } |z| < 1.$$

The Laplace transform of two-parameter Mittag–Leffler function $\mathcal{E}_{\tilde{\alpha},\beta}(z)$ is described using the following integral

$$\int_0^\infty e^{-\mathsf{s}\mathsf{t}}\mathsf{t}^{\beta-1}\mathcal{E}_{\tilde{\alpha},\beta}(\pm a\mathsf{t}^{\tilde{\alpha}})d\mathsf{t} = \frac{\mathsf{s}^{\tilde{\alpha}-\beta}}{(\mathsf{s}^{\tilde{\alpha}}\mp a)}.$$

That is, $\mathcal{L}\{\mathsf{t}^{\beta-1}\mathcal{E}_{\tilde{\alpha},\beta}(\pm a\mathsf{t}^{\tilde{\alpha}})\}(\mathsf{s}) = \frac{\mathsf{s}^{\tilde{\alpha}-\beta}}{(\mathsf{s}^{\tilde{\alpha}}\mp a)}$.

**Lemma 1.** *Kreyszig (1978): Suppose that the bounded linear operator $A_l : \mathbb{R}^{n_l} \to \mathbb{R}^{n_l}$ is determined on a Banach space. Take that $\|A_l\| < 1$. Then $(I - A_l)^{-1}$ is linear and bounded, $(I - A_l)^{-1} = \sum_{i=0}^\infty A_l^i$. Then, $\|(I - A_l)^{-1}\| \leq (1 - \|A_l\|)^{-1}$.*

**Lemma 2.** *Mao (1997): Let $\tilde{g}_l \in \mathcal{M}^2(J;\mathbb{R}^{d\times m}) \ni$*

$$\mathbb{E}\int_0^T |\tilde{\sigma}_l(\mathsf{s})|^p d\mathsf{s} < \infty. \text{ Then, } \mathbb{E}\left|\int_0^T \tilde{\sigma}_l(\mathsf{s})dB(\mathsf{s})\right|^p \leq \left(\frac{p(p-1)}{2}\right)^{\frac{p}{2}} T^{\frac{p-2}{2}}\mathbb{E}\int_0^T |\tilde{\sigma}_l(\mathsf{s})|^p d\mathsf{s}$$

*where $p \geq 2$.*

**Lemma 3.** *Applebaum (2009): For any $p \geq 2$, there exists $\bar{\tilde{A}}_k > 0$, such that*

$$\mathbb{E}\sup_{\mathsf{s}\in[0,\mathsf{t}]}\left\|\int_0^{\mathsf{s}}\int_{-\infty}^{+\infty}\tilde{g}_\mathsf{k}(v,z)\widehat{N}(dv,dz)\right\|^p \leq \bar{\tilde{A}}_k\left\{\mathbb{E}\left[\left(\int_0^{\mathsf{t}}\int_{-\infty}^{+\infty}\|\tilde{g}_\mathsf{k}(\mathsf{s},z)\|^2\kappa(dz)d\mathsf{s}\right)^{\frac{p}{2}}\right]\right.$$
$$\left. + \mathbb{E}\left[\int_0^{\mathsf{t}}\int_{-\infty}^{+\infty}\|\tilde{g}_\mathsf{k}(\mathsf{s},z)\|^p\kappa(dz)d\mathsf{s}\right]\right\}.$$

**Definition 4.** *A normalized fBm $w^{\mathcal{H}} = \{w_{(t)}^{\mathcal{H}} : 0 \leq t < \infty\}$ with $0 < \mathcal{H} < 1$ on $(\Omega, \mathcal{F}, P)$ is uniquely characterized by the following properties:*

- $w_{(t)}^{\mathcal{H}}$ *has stationary increments;*
- $w_{(0)}^{\mathcal{H}} = 0$, *and* $\mathbb{E}w_{(t)}^{\mathcal{H}} = 0$ *for* $t \geq 0$;
- $w_{(t)}^{\mathcal{H}}$ *has a Gaussian distribution for $t > 0$.*

*From the above three properties, it follows that the covariance function is given by*

$$R_{\mathcal{H}}(s,t) = \mathbb{E}\left(w_{(s)}^{\mathcal{H}}w_{(t)}^{\mathcal{H}}\right) = \frac{1}{2}\left\{t^{2\mathcal{H}} + s^{2\mathcal{H}} - |t-s|^{2\mathcal{H}}\right\} \text{ for } 0 < s \leq t.$$

**Definition 5.** *Seemab and Rehman (2018): The solution $\mathsf{x}_l(\mathsf{t}) = \varphi(\mathsf{t})$ of (1) is called stable, if for every $\epsilon > 0$ and $\mathsf{t}_0 \geq 0$, $\exists \delta = \delta(\mathsf{t}_0, \epsilon) > 0 \ni |\mathsf{x}_l(\mathsf{t}, \mathsf{x}_{l0}, \mathsf{t}_0) - \varphi(\mathsf{t})| < \epsilon$ for $|\mathsf{x}_{l0} - \varphi(\mathsf{t}_0)| \leq \delta(\mathsf{t}_0, \epsilon)$ and all $\mathsf{t} \geq \mathsf{t}_0$.*

**Definition 6.** *Equation (1) is said to be exponentially stable if $\exists \mu$ is positive, $1 \leq \mathbb{M}^* \ni \mathsf{t} \geq 0$,*

$$\mathbb{E}\|\mathsf{x}_l(\mathsf{t})\|^2 \leq \mathbb{M}^* e^{-\mu\mathsf{t}}.$$

*The solution of Equation* (1) *can be explained as follows*

$$
\begin{aligned}
x_l(t) =& \mathcal{E}_{\tilde{\alpha}}(\breve{\mathcal{A}}_l t^{\tilde{\alpha}}) \Big[ \varphi(0) + \tilde{g}_l(0, \varphi(0)) \Big] + \tilde{g}_l(t, x_l(t), x_l(t - \tilde{h}(t))) \\
& + \int_0^t (t-s)^{\tilde{\alpha}-1} \mathcal{E}_{\tilde{\alpha},\tilde{\alpha}}(\breve{\mathcal{A}}_l(t-s)^{\tilde{\alpha}}) \tilde{f}_l(s, x_l(s), x_l(s - \tilde{h}(s))) ds \\
& + \int_0^t (t-s)^{\tilde{\alpha}-1} \mathcal{E}_{\tilde{\alpha},\tilde{\alpha}}(\breve{\mathcal{A}}_l(t-s)^{\tilde{\alpha}}) \left[ \int_0^s \tilde{\sigma}_l(\tilde{\tau}, x_l(\tilde{\tau}), x_l(\tilde{\tau} - \tilde{h}(\tilde{\tau}))) dw(\tilde{\tau}) \right] ds \\
& + \int_0^t (t-s)^{\tilde{\alpha}-1} \mathcal{E}_{\tilde{\alpha},\tilde{\alpha}}(\breve{\mathcal{A}}_l(t-s)^{\tilde{\alpha}}) \breve{\mathcal{A}}_l \tilde{g}_l(s, x_l(s), x_l(s - \tilde{h}(s))) ds \\
& + \int_0^t (t-s)^{\tilde{\alpha}-1} \mathcal{E}_{\tilde{\alpha},\tilde{\alpha}}(\breve{\mathcal{A}}_l(t-s)^{\tilde{\alpha}}) \left[ \int_0^s \tilde{\eta}_l(\tilde{\tau}, x_l(\tilde{\tau}), x_l(\tilde{\tau} - \tilde{h}(\tilde{\tau}))) dw_{(\tilde{\tau})}^{\mathcal{H}} \right] ds.
\end{aligned}
$$

## 3. Existence and Uniqueness of Solutions

In this section, we show the existence and uniqueness of solutions and stability results. As a result, we establish the below hypothesis:

$(H_1)$ For $\tilde{f}_l, \tilde{\sigma}_l, \tilde{g}_l \; \exists \; q > 1$ (constant) and $V_{\tilde{f}_l}(\cdot)$, $V_{\tilde{\sigma}_l}(\cdot)$ and $V_{\tilde{g}_l}(\cdot) \in L^q(J, \mathbb{R}^+) \ni$

(i)　$\mathbb{E}\|\tilde{f}_l(t, x_l(t), x_l(t - \tilde{h}(t))) - \tilde{f}_l(t, y_l(t), y_l(t - \tilde{h}(t)))\|^2 \leq V_{\tilde{f}_l}(t) \, \mathbb{E}\|x_l(t) - y_l(t)\|^2$

(ii)　$\mathbb{E}\|\tilde{\sigma}_l(t, x_l(t), x_l(t - \tilde{h}(t))) - \tilde{\sigma}_l(t, y_l(t), y_l(t - \tilde{h}(t)))\|^2 \leq V_{\tilde{\sigma}_l}(t) \, \mathbb{E}\|x_l(t) - y_l(t)\|^2$

(iii)　$\mathbb{E}\|\tilde{g}_l(t, x_l(t), x_l(t - \tilde{h}(t))) - \tilde{g}_l(t, y_l(t), y_l(t - \tilde{h}(t)))\|^2 \leq V_{\tilde{g}_l}(t) \, \mathbb{E}\|x_l(t) - y_l(t)\|^2.$

(iv)　$\mathbb{E}\| \int_0^t \tilde{\eta}_l(\tilde{\tau}, x_l(\tilde{\tau}), x_l(\tilde{\tau} - \tilde{h}(\tilde{\tau}))) dw_{(\tilde{\tau})}^{\mathcal{H}} - \int_0^t \tilde{\eta}_l(\tilde{\tau}, y_l(\tilde{\tau}), y_l(\tilde{\tau} - \tilde{h}(\tilde{\tau}))) dw_{(\tilde{\tau})}^{\mathcal{H}} \|^2$
$\leq 2\mathcal{H} t^{2\mathcal{H}-1} \int_0^s V_{\tilde{\eta}_l}(t) \mathbb{E}\|x_l(t) - y_l(t)\|_{L^2}^2 ds.$

$(H_2)$ The below properties are true, for $t \geq 0$, $N_1, N_2 \geq 1$

(i)　$\|\mathcal{E}_{\tilde{\alpha}}(\breve{\mathcal{A}}_l t^{\tilde{\alpha}})\| \leq N_1 e^{-\omega t}.$

(ii)　$\|\mathcal{E}_{\tilde{\alpha},\tilde{\alpha}}(\breve{\mathcal{A}}_l(t-s)^{\tilde{\alpha}})\| \leq N_2 \, e^{-\omega(t-s)}.$

$(H_3) \exists \; \widehat{V}_{\tilde{f}_l}, \widehat{V}_{\tilde{\sigma}_l}$ (constants), and $\widehat{V}_{\tilde{g}_l} \ni$

(i)　$\mathbb{E}\|\tilde{f}_l(t, x_l(t), x_l(t - \tilde{h}(t)))\|^2 \leq \widehat{V}_{\tilde{f}_l} (1 + \mathbb{E}\|x_l(t)\|^2)$

(ii)　$\mathbb{E}\|\tilde{\sigma}_l(t, x_l(t), x_l(t - \tilde{h}(t)))\|^2 \leq \widehat{V}_{\tilde{\sigma}_l} (1 + \mathbb{E}\|x_l(t)\|^2)$

(iii)　$\mathbb{E}\|\tilde{g}_l(t, x_l(t), x_l(t - \tilde{h}(t)))\|^2 \leq \widehat{V}_{\tilde{\cdot}_l} (1 + \mathbb{E}\|x_l(t)\|^2).$

(iv)　$\mathbb{E}\| \int_0^t \tilde{\eta}_l(\tilde{\tau}, x_l(\tilde{\tau}), x_l(\tilde{\tau} - \tilde{h}(\tilde{\tau}))) dw_{(\tilde{\tau})}^{\mathcal{H}} \|^2 \leq 2\mathcal{H} t^{2\mathcal{H}-1} \int_0^t V_{\tilde{\eta}_l}(s) \mathbb{E}\|1 + x_l(s)\|_{L^2}^2 ds.$

In addition, we set

$$
\begin{aligned}
Q_1 =& 5\widehat{V}_{\tilde{g}_l} + 10N_2 \left( \frac{1 - e^{-2p\omega T}}{2p\omega} \right)^{\frac{1}{p}} \left[ \frac{T^{2\tilde{\alpha}-1}}{2\tilde{\alpha}-1} \|V_{\tilde{f}_l}\|_{L^q(J,\mathbb{R}^+)} + \frac{T^{2\tilde{\alpha}}}{\tilde{\alpha}^2} \|V_{\tilde{\sigma}_l}\|_{L^q(J,\mathbb{R}^+)} \right. \\
& \left. + \frac{T^{2\tilde{\alpha}-1}}{2\tilde{\alpha}-1} \breve{\mathcal{A}}_l \|V_{\tilde{g}_l}\|_{L^q(J,\mathbb{R}^+)} + 2\mathcal{H} t^{2\mathcal{H}-1} \frac{T^{2\tilde{\alpha}}}{\tilde{\alpha}^2} \|V_{\tilde{\eta}_l}\|_{L^q(J,\mathbb{R}^+)} \right] \\
Q_2 =& 5\widehat{V}_{\tilde{g}_l} + 10N_2 \left( \frac{1 - e^{-2\omega T}}{2\omega} \right) \left[ \frac{T^{2\tilde{\alpha}-1}}{2\tilde{\alpha}-1} R_{\tilde{f}_l} + \frac{T^{2\tilde{\alpha}}}{\tilde{\alpha}^2} R_{\tilde{\sigma}_l} + \frac{T^{2\tilde{\alpha}-1}}{2\tilde{\alpha}-1} \breve{\mathcal{A}}_l R_{\tilde{g}_l} + 2\mathcal{H} t^{2\mathcal{H}-1} \frac{T^{2\tilde{\alpha}}}{\tilde{\alpha}^2} R_{\tilde{\eta}_l} \right].
\end{aligned}
$$

Here, we take $R_{\tilde{f}_l} = \sup\limits_{t\in J} \mathbb{E}\|\tilde{f}_l(t,0,0)\|^2, R_{\tilde{\sigma}_l} = \sup\limits_{t\in J} \mathbb{E}\|\tilde{\sigma}_l(t,0,0)\|^2, R_{\tilde{g}_l} = \sup\limits_{t\in J} \mathbb{E}\|\tilde{g}_l(t,0,0)\|^2$ and $R_{\tilde{\eta}_l} = \sup\limits_{t\in J} \mathbb{E}\|\tilde{\eta}_l(t,0,0)\|^2$.

**Theorem 1.** *Consider hypothesis $(H_1)$ and $(H_2)$ are true; then (1) has at least one solution provided that*

$$
\begin{aligned}
M_2 := & 4V_{\tilde{g}_l} + 4N_2\left(\frac{1-e^{-2p\omega T}}{2p\omega}\right)^{\frac{1}{p}}\left[\frac{T^{2\tilde{\alpha}-1}}{2\tilde{\alpha}-1}\|V_{\tilde{f}_l}\|_{L^q(J,\mathbb{R}^+)} + \frac{T^{2\tilde{\alpha}}}{\tilde{\alpha}^2}\|V_{\tilde{\sigma}_l}\|_{L^q(J,\mathbb{R}^+)}\right. \\
& \left. + \frac{T^{2\tilde{\alpha}-1}}{2\tilde{\alpha}-1}\bar{\mathcal{A}}_l\|V_{\tilde{g}_l}\|_{L^q(J,\mathbb{R}^+)} + 2\mathcal{H}t^{2\mathcal{H}-1}\frac{T^{2\tilde{\alpha}}}{\tilde{\alpha}^2}\|V_{\tilde{\eta}_l}\|_{L^q(J,\mathbb{R}^+)}\right] < 1,
\end{aligned}
\tag{2}
$$

*where $\frac{1}{p}+\frac{1}{q} = 1$, $p,q > 1$ and $\mathsf{x}_l \equiv 0$ (the trivial solution) of Equation (1) are stable in $\mathcal{B}$.*

**Proof.** For each $r \geq 0$, define $\mathcal{B}_r = \{\mathsf{x}_l(\mathsf{t}) : \mathsf{x}_l(\mathsf{t}) \in \mathcal{B}; \mathbb{E}\|\mathsf{x}_l(\mathsf{t})\|^2 \leq r\}$ and then for each $r$, $\mathcal{B}_r$ is a bounded, closed and convex subset of $\mathcal{B}$. Define the operator $\Phi : \mathcal{B}_r \to \mathcal{B}_r$

$$
\begin{aligned}
(\Phi\mathsf{x}_l)(\mathsf{t}) = & \mathcal{E}_{\tilde{\alpha}}(\bar{\mathcal{A}}_l\mathsf{t}^{\tilde{\alpha}})\Big[\varphi(0) + \tilde{g}_l(0,\varphi(0))\Big] + \tilde{g}_l(\mathsf{t},\mathsf{x}_l(\mathsf{t}),\mathsf{x}_l(\mathsf{t}-\tilde{h}(\mathsf{t}))) \\
& + \int_0^{\mathsf{t}} (\mathsf{t}-\mathsf{s})^{\tilde{\alpha}-1}\mathcal{E}_{\tilde{\alpha},\tilde{\alpha}}(\bar{\mathcal{A}}_l(\mathsf{t}-\mathsf{s})^{\tilde{\alpha}})\tilde{f}_l(\mathsf{s},\mathsf{x}_l(\mathsf{s}),\mathsf{x}_l(\mathsf{s}-\tilde{h}(\mathsf{s})))d\mathsf{s} \\
& + \int_0^{\mathsf{t}} (\mathsf{t}-\mathsf{s})^{\tilde{\alpha}-1}\mathcal{E}_{\tilde{\alpha},\tilde{\alpha}}(\bar{\mathcal{A}}_l(\mathsf{t}-\mathsf{s})^{\tilde{\alpha}})\left[\int_0^{\mathsf{s}} \tilde{\sigma}_l(\tilde{\tau},\mathsf{x}_l(\tilde{\tau}),\mathsf{x}_l(\tilde{\tau}-\tilde{h}(\tilde{\tau})))dw(\tilde{\tau})\right]d\mathsf{s} \\
& + \int_0^{\mathsf{t}} (\mathsf{t}-\mathsf{s})^{\tilde{\alpha}-1}\mathcal{E}_{\tilde{\alpha},\tilde{\alpha}}(\bar{\mathcal{A}}_l(\mathsf{t}-\mathsf{s})^{\tilde{\alpha}})\bar{\mathcal{A}}_l\tilde{g}_l(\mathsf{s},\mathsf{x}_l(\mathsf{s}),\mathsf{x}_l(\mathsf{s}-\tilde{h}(\mathsf{s})))d\mathsf{s} \\
& + \int_0^{\mathsf{t}} (\mathsf{t}-\mathsf{s})^{\tilde{\alpha}-1}\mathcal{E}_{\tilde{\alpha},\tilde{\alpha}}(\bar{\mathcal{A}}_l(\mathsf{t}-\mathsf{s})^{\tilde{\alpha}})\left[\int_0^{\mathsf{s}} \tilde{\eta}_l(\tilde{\tau},\mathsf{x}_l(\tilde{\tau}),\mathsf{x}_l(\tilde{\tau}-\tilde{h}(\tilde{\tau})))dw_{(\tilde{\tau})}^{\mathcal{H}}\right]d\mathsf{s}.
\end{aligned}
$$

**Step I:** To prove that $\exists\, r \geq 0 \ni \Phi(\mathcal{B}_r) \subseteq \mathcal{B}_r$. Based on $(H_1)$, $(H_2)$ and Hölder inequality, we get

$$
\mathbb{E}\left\|\int_0^{\mathsf{t}} (\mathsf{t}-\mathsf{s})^{\tilde{\alpha}-1}\mathcal{E}_{\tilde{\alpha},\tilde{\alpha}}(\bar{\mathcal{A}}_l(\mathsf{t}-\mathsf{s})^{\tilde{\alpha}})\tilde{f}_l(\mathsf{s},\mathsf{x}_l(\mathsf{s}),\mathsf{x}_l(\mathsf{s}-\tilde{h}(\mathsf{s})))d\mathsf{s}\right\|^2
$$

$$
\leq \frac{T^{2\tilde{\alpha}-1}}{2\tilde{\alpha}-1}N_2\int_0^{\mathsf{t}} e^{-2\omega(\mathsf{t}-\mathsf{s})}\mathbb{E}\|\tilde{f}_l(\mathsf{s},\mathsf{x}_l(\mathsf{s}),\mathsf{x}_l(\mathsf{s}-\tilde{h}(\mathsf{s}))) - \tilde{f}_l(\mathsf{s},0,0) + \tilde{f}_l(\mathsf{s},0,0)\|^2 d\mathsf{s}
$$

$$
\leq 2\frac{T^{2\tilde{\alpha}-1}}{2\tilde{\alpha}-1}N_2\left\{\int_0^{\mathsf{t}} e^{-2\omega(\mathsf{t}-\mathsf{s})}V_{\tilde{f}_l}(\mathsf{s})\mathbb{E}\|\mathsf{x}_l(\mathsf{s})\|^2 d\mathsf{s} + \int_0^{\mathsf{t}} e^{-2\omega(\mathsf{t}-\mathsf{s})}\mathbb{E}\|\tilde{f}_l(\mathsf{s},0,0)\|^2 d\mathsf{s}\right\}
$$

$$
\leq 2\frac{T^{2\tilde{\alpha}-1}}{2\tilde{\alpha}-1}N_2\left\{\left(\int_0^{\mathsf{t}} e^{-2p\omega(\mathsf{t}-\mathsf{s})}d\mathsf{s}\right)^{\frac{1}{p}}\left(\int_0^{\mathsf{t}} V_{\tilde{f}_l}^q(\mathsf{s})d\mathsf{s}\right)^{\frac{1}{q}}\mathbb{E}\|\mathsf{x}_l\|^2 + R_{\tilde{f}_l}\int_0^{\mathsf{t}} e^{-2\omega(\mathsf{t}-\mathsf{s})}d\mathsf{s}\right\}
$$

$$
\leq 2\frac{T^{2\tilde{\alpha}-1}}{2\tilde{\alpha}-1}N_2\left\{\left(\frac{1-e^{-2p\omega T}}{2p\omega}\right)^{\frac{1}{p}}\|V_{\tilde{f}_l}\|_{L^q(J,\mathbb{R}^+)}r + R_{\tilde{f}_l}\left(\frac{1-e^{-2\omega T}}{2\omega}\right)\right\}.
$$

Similarly,

$$\mathbb{E}\left\|\int_0^t (t-s)^{\tilde{\alpha}-1} \mathcal{E}_{\tilde{\alpha},\tilde{\alpha}}(\bar{\mathcal{A}}_l(t-s)^{\tilde{\alpha}})\left[\int_0^s \tilde{\sigma}_l(\tilde{\tau}, x_l(\tilde{\tau}), x_l(\tilde{\tau}-\tilde{h}(\tilde{\tau})))dw(\tilde{\tau})\right]ds\right\|^2$$

$$\leq 2\frac{T^{2\tilde{\alpha}}}{\tilde{\alpha}^2}N_2\left\{\left(\frac{1-e^{-2p\omega T}}{2p\omega}\right)^{\frac{1}{p}}\|V_{\tilde{\sigma}_l}\|_{L^q(J,\mathbb{R}^+)}r + R_{\tilde{\sigma}_l}\left(\frac{1-e^{-2\omega T}}{2\omega}\right)\right\},$$

$$\mathbb{E}\left\|\int_0^t (t-s)^{\tilde{\alpha}-1} \mathcal{E}_{\tilde{\alpha},\tilde{\alpha}}(\bar{\mathcal{A}}_l(t-s)^{\tilde{\alpha}})\bar{\mathcal{A}}_l\tilde{g}_l(s, x_l(s), x_l(s-\tilde{h}(s)))ds\right\|^2$$

$$\leq 2\frac{T^{2\tilde{\alpha}-1}}{2\tilde{\alpha}-1}N_2\bar{\mathcal{A}}_l\left\{\left(\frac{1-e^{-2p\omega T}}{2p\omega}\right)^{\frac{1}{p}}\|V_{\tilde{g}_l}\|_{L^q(J,\mathbb{R}^+)}r + R_{\tilde{g}_l}\left(\frac{1-e^{-2\omega T}}{2\omega}\right)\right\}$$

and

$$\mathbb{E}\left\|\int_0^t (t-s)^{\tilde{\alpha}-1} \mathcal{E}_{\tilde{\alpha},\tilde{\alpha}}(\bar{\mathcal{A}}_l(t-s)^{\tilde{\alpha}})\left[\int_0^s \tilde{\eta}_l(\tilde{\tau}, x_l(\tilde{\tau}), x_l(\tilde{\tau}-\tilde{h}(\tilde{\tau})))dw_{(\tilde{\tau})}^{\mathcal{H}}\right]ds\right\|^2$$

$$\leq 4\mathcal{H}t^{2\mathcal{H}-1}\frac{T^{2\tilde{\alpha}}}{\tilde{\alpha}^2}N_2\left\{\left(\frac{1-e^{-2p\omega T}}{2p\omega}\right)^{\frac{1}{p}}\|V_{\tilde{\eta}_l}\|_{L^q(J,\mathbb{R}^+)}r + R_{\tilde{\eta}_l}\left(\frac{1-e^{-2\omega T}}{2\omega}\right)\right\}$$

Now,

$$\mathbb{E}\|(\Phi x_l)(t)\|^2 \leq 5\left\{\mathbb{E}\|\mathcal{E}_{\tilde{\alpha}}(\bar{\mathcal{A}}_l t^{\tilde{\alpha}})[\varphi(0)+\tilde{g}_l(0,\varphi(0))]\|^2 + \mathbb{E}\|\tilde{g}_l(t, x_l(t), x_l(t-\tilde{h}(t)))\|^2\right.$$

$$+\mathbb{E}\left\|\int_0^t (t-s)^{\tilde{\alpha}-1} \mathcal{E}_{\tilde{\alpha},\tilde{\alpha}}(\bar{\mathcal{A}}_l(t-s)^{\tilde{\alpha}})\tilde{f}_l(s, x_l(s), x_l(s-\tilde{h}(s)))ds\right\|^2$$

$$+\mathbb{E}\left\|\int_0^t (t-s)^{\tilde{\alpha}-1} \mathcal{E}_{\tilde{\alpha},\tilde{\alpha}}(\bar{\mathcal{A}}_l(t-s)^{\tilde{\alpha}})\left[\int_0^s \tilde{\sigma}_l(\tilde{\tau}, x_l(\tilde{\tau}), x_l(\tilde{\tau}-\tilde{h}(\tilde{\tau})))dw(\tilde{\tau})\right]ds\right\|^2$$

$$+\mathbb{E}\left\|\int_0^t (t-s)^{\tilde{\alpha}-1} \mathcal{E}_{\tilde{\alpha},\tilde{\alpha}}(\bar{\mathcal{A}}_l(t-s)^{\tilde{\alpha}})\bar{\mathcal{A}}_l\tilde{g}_l(s, x_l(s), x_l(s-\tilde{h}(s)))ds\right\|^2$$

$$+\mathbb{E}\left\|\int_0^t (t-s)^{\tilde{\alpha}-1} \mathcal{E}_{\tilde{\alpha},\tilde{\alpha}}(\bar{\mathcal{A}}_l(t-s)^{\tilde{\alpha}})\left[\int_0^s \tilde{\eta}_l(\tilde{\tau}, x_l(\tilde{\tau}), x_l(\tilde{\tau}-\tilde{h}(\tilde{\tau})))dw_{(\tilde{\tau})}^{\mathcal{H}}\right]ds\right\|^2\right\}$$

$$\leq 5\left\{N_1 e^{-2\omega T}\mathbb{E}\|\varphi(0)+\tilde{g}_l(0,\varphi(0))\|^2 + \widehat{V}_{\tilde{g}_l}\left(1+\mathbb{E}\|x_l(t)\|^2\right)\right.$$

$$+2\frac{T^{2\tilde{\alpha}-1}}{2\tilde{\alpha}-1}N_2\left[\left(\frac{1-e^{-2p\omega T}}{2p\omega}\right)^{\frac{1}{p}}\frac{T^{2\tilde{\alpha}-1}}{2\tilde{\alpha}-1}r + R_{\tilde{f}_l}\left(\frac{1-e^{-2\omega T}}{2\omega}\right)\right]$$

$$+2\frac{T^{2\tilde{\alpha}}}{\tilde{\alpha}^2}N_2\left[\left(\frac{1-e^{-2p\omega T}}{2p\omega}\right)^{\frac{1}{p}}\|V_{\tilde{\sigma}_l}\|_{L^q(J,\mathbb{R}^+)}r + R_{\tilde{\sigma}_l}\left(\frac{1-e^{-2\omega T}}{2\omega}\right)\right]$$

$$
+ 2\frac{T^{2\tilde{\alpha}-1}}{2\tilde{\alpha}-1} N_2 \bar{\mathcal{A}}_l \left[ \left( \frac{1-e^{-2p\omega T}}{2p\omega} \right)^{\frac{1}{p}} \|V_{\tilde{g}_l}\|_{L^q(J,\mathbb{R}^+)} r + R_{\tilde{g}_l} \left( \frac{1-e^{-2\omega T}}{2\omega} \right) \right]
$$

$$
+ 4\mathcal{H}\mathsf{t}^{2\mathcal{H}-1} \frac{T^{2\tilde{\alpha}}}{\tilde{\alpha}^2} N_2 \left\{ \left( \frac{1-e^{-2p\omega T}}{2p\omega} \right)^{\frac{1}{p}} \|V_{\tilde{\eta}_l}\|_{L^q(J,\mathbb{R}^+)} r + R_{\tilde{\eta}_l} \left( \frac{1-e^{-2\omega T}}{2\omega} \right) \right\} \Bigg\} \Bigg\}
$$

$$
\leq 5N_1 e^{-2\omega T} \mathbb{E}\|\varphi(0) + \tilde{g}_l(0, \varphi(0))\|^2 + 5\widehat{V}_{\tilde{g}_l} + 10N_2 \Bigg\{ \left( \frac{1-e^{-2p\omega T}}{2p\omega} \right)^{\frac{1}{p}}
$$

$$
\times \left[ \frac{T^{2\tilde{\alpha}-1}}{2\tilde{\alpha}-1} \|V_{\tilde{f}_l}\|_{L^q(J,\mathbb{R}^+)} + \frac{T^{2\tilde{\alpha}}}{\tilde{\alpha}^2} \|V_{\tilde{\sigma}_l}\|_{L^q(J,\mathbb{R}^+)} + \frac{T^{2\tilde{\alpha}-1}}{2\tilde{\alpha}-1} \bar{\mathcal{A}}_l \|V_{\tilde{g}_l}\|_{L^q(J,\mathbb{R}^+)} \right.
$$

$$
\left. + 2\mathcal{H}\mathsf{t}^{2\mathcal{H}-1} \frac{T^{2\tilde{\alpha}}}{\tilde{\alpha}^2} \|V_{\tilde{\eta}_l}\|_{L^q(J,\mathbb{R}^+)} + \right] \Bigg\} \Bigg\} r
$$

$$
+ 5\widehat{V}_{\tilde{g}_l} + 10N_2 \left( \frac{1-e^{-2\omega T}}{2\omega} \right) \left[ \frac{T^{2\tilde{\alpha}-1}}{2\tilde{\alpha}-1} R_{\tilde{f}_l} + \frac{T^{2\tilde{\alpha}}}{\tilde{\alpha}^2} R_{\tilde{\sigma}_l} + \frac{T^{2\tilde{\alpha}-1}}{2\tilde{\alpha}-1} \bar{\mathcal{A}}_l R_{\tilde{g}_l} + 2\mathcal{H}\mathsf{t}^{2\mathcal{H}-1} \frac{T^{2\tilde{\alpha}}}{\tilde{\alpha}^2} R_{\tilde{\eta}_l} \right]
$$

$$
\leq 5N_1 e^{-2\omega T} \mathbb{E}\|\varphi(0) + \tilde{g}_l(0, \varphi(0))\|^2 + Q_2 + Q_1 r = r.
$$

For, $r = \frac{5N_1 e^{-2\omega T} \mathbb{E}\|\varphi(0) + \tilde{g}_l(0,\varphi(0))\|^2 + Q_2}{(1-Q_1)}$, $Q_1 < 1$. Hence, we obtain $\Phi(\mathcal{B}_r) \subseteq \mathcal{B}_r$ for such an $r$.

**Step II.** To prove that $\Phi$ is a contraction.

Assume $x_l, y_l \in \mathcal{B}_r$. Using, $(H_1), (H_2)$ and Hölder inequality, for every $\mathsf{t} \in J$, we get

$$
\mathbb{E}\|(\Phi x_l)(\mathsf{t}) - (\Phi y_l)(\mathsf{t})\|^2
$$

$$
= \mathbb{E}\Bigg\{ \Bigg\| \mathcal{E}_{\tilde{\alpha}}(\bar{\mathcal{A}}_l \mathsf{t}^{\tilde{\alpha}})[\varphi(0) + \tilde{g}_l(0,\varphi(0))] + \tilde{g}_l(s, x_l(s), x_l(s - \tilde{h}(s)))
$$

$$
+ \int_0^{\mathsf{t}} (\mathsf{t}-s)^{\tilde{\alpha}-1} \mathcal{E}_{\tilde{\alpha},\tilde{\alpha}}(\bar{\mathcal{A}}_l(\mathsf{t}-s)^{\tilde{\alpha}}) \tilde{f}_l(s, x_l(s), x_l(s - \tilde{h}(s))) ds
$$

$$
+ \int_0^{\mathsf{t}} (\mathsf{t}-s)^{\tilde{\alpha}-1} \mathcal{E}_{\tilde{\alpha},\tilde{\alpha}}(\bar{\mathcal{A}}_l(\mathsf{t}-s)^{\tilde{\alpha}}) \left[ \int_0^s \tilde{\sigma}_l(\tilde{\tau}, x_l(\tilde{\tau}), x_l(\tilde{\tau} - \tilde{h}(\tilde{\tau}))) dw(\tilde{\tau}) \right] ds
$$

$$
+ \int_0^{\mathsf{t}} (\mathsf{t}-s)^{\tilde{\alpha}-1} \mathcal{E}_{\tilde{\alpha},\tilde{\alpha}}(\bar{\mathcal{A}}_l(\mathsf{t}-s)^{\tilde{\alpha}}) \bar{\mathcal{A}}_l \tilde{g}_l(s, x_l(s), x_l(s - \tilde{h}(s))) ds
$$

$$
+ \int_0^{\mathsf{t}} (\mathsf{t}-s)^{\tilde{\alpha}-1} \mathcal{E}_{\tilde{\alpha},\tilde{\alpha}}(\bar{\mathcal{A}}_l(\mathsf{t}-s)^{\tilde{\alpha}}) \left[ \int_0^s \tilde{\eta}_l(\tilde{\tau}, x_l(\tilde{\tau}), x_l(\tilde{\tau} - \tilde{h}(\tilde{\tau}))) dw^{\mathcal{H}}_{(\tilde{\tau})} \right] ds
$$

$$
- \mathcal{E}_{\tilde{\alpha}}(\bar{\mathcal{A}}_l \mathsf{t}^{\tilde{\alpha}})[\varphi(0) + \tilde{g}_l(0,\varphi(0))] - \tilde{g}_l(s, y_l(s), y_l(s - \tilde{h}(s)))
$$

$$
- \int_0^{\mathsf{t}} (\mathsf{t}-s)^{\tilde{\alpha}-1} \mathcal{E}_{\tilde{\alpha},\tilde{\alpha}}(\bar{\mathcal{A}}_l(\mathsf{t}-s)^{\tilde{\alpha}}) \tilde{f}_l(s, y_l(s), y_l(s - \tilde{h}(s))) ds
$$

$$
- \int_0^{\mathsf{t}} (\mathsf{t}-s)^{\tilde{\alpha}-1} \mathcal{E}_{\tilde{\alpha},\tilde{\alpha}}(\bar{\mathcal{A}}_l(\mathsf{t}-s)^{\tilde{\alpha}}) \left[ \int_0^s \tilde{\sigma}_l(\tilde{\tau}, y_l(\tilde{\tau}), y_l(\tilde{\tau} - \tilde{h}(\tilde{\tau}))) dw(\tilde{\tau}) \right] ds
$$

$$
- \int_0^{\mathsf{t}} (\mathsf{t}-s)^{\tilde{\alpha}-1} \mathcal{E}_{\tilde{\alpha},\tilde{\alpha}}(\bar{\mathcal{A}}_l(\mathsf{t}-s)^{\tilde{\alpha}}) \bar{\mathcal{A}}_l \tilde{g}_l(s, y_l(s), y_l(s - \tilde{h}(s))) ds
$$

$$
- \int_0^t (t-s)^{\tilde{\alpha}-1} \mathcal{E}_{\tilde{\alpha},\tilde{\alpha}}(\bar{\tilde{\mathcal{A}}}_l(t-s)^{\tilde{\alpha}}) \left[ \int_0^s \tilde{\eta}_l(\tilde{\tau}, y_l(\tilde{\tau}), y_l(\tilde{\tau}-\tilde{h}(\tilde{\tau}))) dw_{(\tilde{\tau})}^{\mathcal{H}} \right] ds \Bigg\|^2 \Bigg\}
$$

$$
\leq 4 \Bigg\{ \mathbb{E} \big\| \tilde{g}_l(s, x(s), x(s-\tilde{h}(s))) - \tilde{g}_l(s, y_l(s), y_l(s-\tilde{h}(s))) \big\|^2
$$

$$
+ \mathbb{E} \left\| \int_0^t (t-s)^{\tilde{\alpha}-1} \mathcal{E}_{\tilde{\alpha},\tilde{\alpha}}(\bar{\tilde{\mathcal{A}}}_l(t-s)^{\tilde{\alpha}}) \left[ \tilde{f}_l(s, x_l(s), x_l(s-\tilde{h}(s))) - \tilde{f}_l(s, y_l(s), y_l(s-\tilde{h}(s))) \right] ds \right\|^2
$$

$$
+ \mathbb{E} \left\| \int_0^t (t-s)^{\tilde{\alpha}-1} \mathcal{E}_{\tilde{\alpha},\tilde{\alpha}}(\bar{\tilde{\mathcal{A}}}_l(t-s)^{\tilde{\alpha}}) \right.
$$

$$
\times \left[ \int_0^s \Big( \tilde{\sigma}_l(\tilde{\tau}, x_l(\tilde{\tau}), x_l(\tilde{\tau}-\tilde{h}(\tilde{\tau}))) - \tilde{\sigma}_l(\tilde{\tau}, y_l(\tilde{\tau}), y_l(\tilde{\tau}-\tilde{h}(\tilde{\tau}))) \Big) dw(\tilde{\tau}) \right] ds \Bigg\|^2
$$

$$
+ \mathbb{E} \left\| \int_0^t (t-s)^{\tilde{\alpha}-1} \mathcal{E}_{\tilde{\alpha},\tilde{\alpha}}(\bar{\tilde{\mathcal{A}}}_l(t-s)^{\tilde{\alpha}}) \bar{\tilde{\mathcal{A}}}_l \right.
$$

$$
\times \Big( \tilde{g}_l(s, x_l(s), x_l(s-\tilde{h}(s))) - \tilde{g}_l(s, y_l(s), y_l(s-\tilde{h}(s))) \Big) ds
$$

$$
+ \int_0^t (t-s)^{\tilde{\alpha}-1} \mathcal{E}_{\tilde{\alpha},\tilde{\alpha}}(\bar{\tilde{\mathcal{A}}}_l(t-s)^{\tilde{\alpha}})
$$

$$
\times \left[ \int_0^s \Big( \tilde{\eta}_l(\tilde{\tau}, x_l(\tilde{\tau}), x_l(\tilde{\tau}-\tilde{h}(\tilde{\tau}))) - \tilde{\eta}_l(\tilde{\tau}, y_l(\tilde{\tau}), y_l(\tilde{\tau}-\tilde{h}(\tilde{\tau}))) \Big) dw_{(\tilde{\tau})}^{\mathcal{H}} \right] ds \Bigg\|^2 \Bigg\}
$$

$$
\leq 4 \|V_{\tilde{g}_l}\|_{L^q(J,\mathbb{R}^+)} + 4N_2 \left( \frac{1-e^{-2p\omega T}}{2p\omega} \right)^{\frac{1}{p}} \left[ \frac{T^{2\tilde{\alpha}-1}}{2\tilde{\alpha}-1} \|V_{\tilde{f}_l}\|_{L^q(J,\mathbb{R}^+)} + \frac{T^{2\tilde{\alpha}}}{\tilde{\alpha}^2} \|V_{\tilde{\sigma}_l}\|_{L^q(J,\mathbb{R}^+)} \right.
$$

$$
+ \frac{T^{2\tilde{\alpha}-1}}{2\tilde{\alpha}-1} \bar{\tilde{\mathcal{A}}}_l \|V_{\tilde{g}_l}\|_{L^q(J,\mathbb{R}^+)} + 2\mathcal{H}t^{2\mathcal{H}-1} \frac{T^{2\tilde{\alpha}}}{\tilde{\alpha}^2} \|V_{\tilde{\eta}_l}\|_{L^q(J,\mathbb{R}^+)} \right] \mathbb{E}\|x_l(t) - y_l(t)\|^2,
$$

which reveals that

$$
\mathbb{E}\|(\Phi x_l)(t) - (\Phi y_l)(t)\|^2 \leq M_2 \mathbb{E}\|x_l - y_l\|^2.
$$

Using (2), we conclude that $M_2 < 1$, which implies $\Phi$ is a contraction mapping with a unique fixed point $x_l(t) \in \mathcal{B}_r$, which is a solution of (1). Now, we prove the stability conditions of (1)

For any given $\varepsilon > 0$, $\exists \lambda = \frac{\varepsilon(1-Q_1)-Q_2}{5N_1 e^{-2\omega T}} \ni \|\varphi(0) + \tilde{g}_l(0, \varphi(0))\|^2 \leq \lambda$, which implies

$$\mathbb{E}\|x_l(t)\|^2 \leq 5N_1 e^{-2\omega T}\mathbb{E}\|\varphi(0)+\tilde{g}_l(0,\varphi(0))\|^2 + 10N_2\left\{\left(\frac{1-e^{-2p\omega T}}{2p\omega}\right)^{\frac{1}{p}}\left[\frac{T^{2\tilde{\alpha}-1}}{2\tilde{\alpha}-1}\|V_{\tilde{f}_l}\|_{L^q(J,\mathbb{R}^+)}\right.\right.$$

$$\left.+\frac{T^{2\tilde{\alpha}}}{\tilde{\alpha}^2}\|V_{\tilde{\sigma}_l}\|_{L^q(J,\mathbb{R}^+)}+\frac{T^{2\tilde{\alpha}-1}}{2\tilde{\alpha}-1}\breve{\mathcal{A}}_l\|V_{\tilde{g}_l}\|_{L^q(J,\mathbb{R}^+)}+2\mathcal{H}t^{2\mathcal{H}-1}\frac{T^{2\tilde{\alpha}}}{\tilde{\alpha}^2}\|V_{\tilde{\eta}_l}\|_{L^q(J,\mathbb{R}^+)}\right]\right\}r$$

$$+10N_2\left(\frac{1-e^{-2\omega T}}{2\omega}\right)\left[\frac{T^{2\tilde{\alpha}-1}}{2\tilde{\alpha}-1}R_{\tilde{f}_l}+\frac{T^{2\tilde{\alpha}}}{\tilde{\alpha}^2}R_{\tilde{\sigma}_l}+\frac{T^{2\tilde{\alpha}-1}}{2\tilde{\alpha}-1}\breve{\mathcal{A}}_l R_{\tilde{g}_l}+2\mathcal{H}t^{2\mathcal{H}-1}\frac{T^{2\tilde{\alpha}}}{\tilde{\alpha}^2}R_{\tilde{\eta}_l}\right]$$

$$\leq 5N_1 e^{-2\omega T}\lambda + Q_1 r + Q_2$$

$$r(1-Q_1)\leq 5N_1 e^{-2\omega T}\lambda + Q_2$$

$$r\leq\epsilon.$$

Thus, the proof is over. □

### 4. Exponential Stability

**Theorem 2.** *If hypotheses* $(H_2)-(H_3)$ *are true, then* (1) *is exponentially stable, provided that*

$$\omega > \beta = N_2\left[\frac{T^{2\tilde{\alpha}-1}}{2\tilde{\alpha}-1}\left(\widehat{V}_{\tilde{f}_l}+\breve{\mathcal{A}}_l\widehat{V}_{\tilde{g}_l}\right)(1+r)+\frac{T^{2\tilde{\alpha}}}{\tilde{\alpha}^2}\widehat{V}_{\tilde{\sigma}_l}(1+r)+2\mathcal{H}t^{2\mathcal{H}-1}\frac{T^{2\tilde{\alpha}}}{\tilde{\alpha}^2}\widehat{V}_{\tilde{\eta}_l}(1+r)\right].\tag{3}$$

**Proof.**

$$\mathbb{E}\|x_l(t)\|^2 \leq 5e^{-2\omega t}N_1\mathbb{E}\|\varphi(0)+\tilde{g}_l(0,\varphi(0))\|^2 + 5N_2 e^{-2\omega t}\left[\frac{T^{2\tilde{\alpha}-1}}{2\tilde{\alpha}-1}(1+r)\left[\widehat{V}_{\tilde{f}_l}+\breve{\mathcal{A}}_l\widehat{V}_{\tilde{g}_l}\right]\right.$$

$$\left.+\frac{T^{2\tilde{\alpha}}}{\tilde{\alpha}^2}\widehat{V}_{\tilde{\sigma}_l}(1+r)+2\mathcal{H}t^{2\mathcal{H}-1}\frac{T^{2\tilde{\alpha}}}{\tilde{\alpha}^2}\widehat{V}_{\tilde{\eta}_l}(1+r)\right]\int_0^t e^{2\omega s}ds$$

$$\mathbb{E}\|x_l(t)\|^2 e^{2\omega t} \leq 5N_1\mathbb{E}\|\varphi(0)+\tilde{g}_l(0,\varphi(0))\|^2 + 5N_2\left[\frac{T^{2\tilde{\alpha}-1}}{2\tilde{\alpha}-1}(1+r)\left[\widehat{V}_{\tilde{f}_l}+\breve{\mathcal{A}}_l\widehat{V}_{\tilde{g}_l}\right]\right.$$

$$\left.+\frac{T^{2\tilde{\alpha}}}{\tilde{\alpha}^2}\widehat{V}_{\tilde{\sigma}_l}(1+r)+2\mathcal{H}t^{2\mathcal{H}-1}\frac{T^{2\tilde{\alpha}}}{\tilde{\alpha}^2}\widehat{V}_{\tilde{\eta}_l}(1+r)\right]\int_0^t e^{2\omega s}ds.$$

We get the result by using the Gronwall's inequality

$$e^{2\omega t}\mathbb{E}\|x_l(t)\|^2 \leq 5N_1\mathbb{E}\|\varphi(0)+\tilde{g}_l(0,\varphi(0))\|^2$$

$$\times\exp\left(5N_2\left[\left(\widehat{V}_{\tilde{f}_l}+\breve{\mathcal{A}}_l\widehat{V}_{\tilde{g}_l}\right)(1+r)+\frac{T^{2\tilde{\alpha}}}{\tilde{\alpha}^2}\widehat{V}_{\tilde{\sigma}_l}(1+r)+2\mathcal{H}t^{2\mathcal{H}-1}\frac{T^{2\tilde{\alpha}}}{\tilde{\alpha}^2}\widehat{V}_{\tilde{\eta}_l}(1+r)\right]t\right).$$

Therefore,

$$\mathbb{E}\|x_l(t)\|^2 \leq M\mathbb{E}\|\varphi(0)+\tilde{g}_l(0,\varphi(0))\|^2\exp((-vt)).$$

where $v = 2\omega - 5\beta$, $M = 5N_1$. Thus, according to (3), (1) is exponentially stable in $\mathcal{B}$. Thus, the proof is over. □

**Remark 1.** *Existence, uniqueness, and stability of mild solutions for second-order neutral stochastic evolution equations with infinite delay and Poisson jumps by the authors in* Ren and Sakthivel *(2012) using successive approximation techniques. The uniqueness and existence of solutions, in*

*addition to their controllability (relative), have been demonstrated using the fixed point approach in Sathiyaraj and Balasubramaniam (2016). In Wang et al. (2017), the authors investigate the controllability of a differential delay semilinear system with linear sections determined by matrices (permutable). We proposed a new real concept of stability results in finite dimensional space in this study by using weaker conditions for nonlinear terms.*

## 5. Numerical Simulations

Consider the system of NFSDS described by

$$
{}^{\mathcal{C}}\tilde{\mathcal{D}}^{0.6}[x_{l1}(t) - (-t+2)e^{-t}x_{l1}(t)] = (0.1)x_{l1}(t) - (3-t)\frac{x_{l1}^2(t)}{1-t} - \int_0^t s x_{l1}(s)\sigma_{l1}dB_1 + \int_0^t 3s x_{l1}(s)\eta_{l1}dB_1^{\mathcal{H}} \tag{4}
$$

$$
{}^{\mathcal{C}}\tilde{\mathcal{D}}^{0.6}[x_{l2}(t) - (2-t)x_{l2}(t)e^{-t}] = -(0.1)x_{l2}(t) - (3-t)\frac{x_{l2}^3(t)}{1-t} - \int_0^t s x_{l2}(s)\sigma_{l2}dB_2 + \int_0^t 5s x_{l2}(s)\eta_{l2}dB_2^{\mathcal{H}} \tag{5}
$$

for $t \in J_1 = [0,1]$ and $0.5 < \tilde{\alpha} < 1$. Let us take

$$
\bar{\tilde{\mathcal{A}}}_l = \begin{pmatrix} 0.1 & 0 \\ 0 & -0.1 \end{pmatrix}, \quad \tilde{f}_l(t, x_l(t), x_l(t - \tilde{h}(t))) = \begin{pmatrix} -(3-t)\frac{x_{l1}^2(t)}{1-t} \\ -(3-t)\frac{x_{l2}^3(t)}{1-t} \end{pmatrix},
$$

$$
\tilde{\sigma}_l(t, x_l(t), x_l(t - \tilde{h}(t))) = \begin{pmatrix} -t x_{l1}(t)\sigma_{l1}dB_1 \\ -t x_{l2}(t)\sigma_{l2}dB_2 \end{pmatrix}, \quad \tilde{g}_l(t, x_l(t), x_l(t - \tilde{h}(t))) = \begin{pmatrix} -(2-t)x_{l1}(t)e^{-t} \\ -(2-t)x_{l2}(t)e^{-t} \end{pmatrix},
$$

$$
\tilde{\eta}_l(t, x_l(t), x_l(t - \tilde{h}(t))) = \begin{pmatrix} 3t x_{l1}(t)\eta_{l1}dB_1^{\mathcal{H}} \\ 5t x_{l2}(t)\eta_{l2}dB_2^{\mathcal{H}} \end{pmatrix} \text{ where, } \tilde{h} = 0.01, \; \sigma_{l1} = 0.3, \; \sigma_{l2} = 0.5 \text{ and }
$$

$\tilde{\alpha} = 0.6$.

Furthermore, it is easy to verify that for any $x_l(t), y_l(t) \in \mathbb{R}^2$.

(i). $\mathbb{E}\|\tilde{f}_l(t, x_k(t), x_l(t - \tilde{h}(t))) - \tilde{f}_l(t, y_l(t), y_l(t - \tilde{h}(t)))\|^2 \leq -(3-t)\mathbb{E}\|x_l(t) - y_l(t)\|^2$

(ii). $\mathbb{E}\|\tilde{\sigma}_l(t, x_l(t), x_l(t - \tilde{h}(t))) - \tilde{\sigma}_l(t, y_l(t), y_l(t - \tilde{h}(t)))\|^2 \leq -0.5t \, \mathbb{E}\|x_l(t) - y_l(t)\|^2$

(iii). $\mathbb{E}\|\tilde{g}_l(t, x_l(t), x_l(t - \tilde{h}(t))) - \tilde{g}_l(t, y_l(t), y_l(t - \tilde{h}(t)))\|^2 \leq -(2-t)\mathbb{E}\|x_l(t) - y_l(t)\|^2$

(iv). $\mathbb{E}\|\tilde{\eta}_l(t, x_l(t), x_l(t - \tilde{h}(t))) - \tilde{\eta}_l(t, y_l(t), y_l(t - \tilde{h}(t)))\|^2 \leq 4t\mathbb{E}\|x_l(t) - y_l(t)\|^2.$

Thus, $\tilde{f}_l, \tilde{\sigma}_l$ and $\tilde{g}_l$ satisfies the assumption $(H_1)$, where we set $V_{\tilde{f}_l}(\cdot), V_{\tilde{\sigma}_l}(\cdot), V_{\tilde{g}_l}(\cdot) \in L^q(J_1, \mathbb{R}^+)$.

Hence, all the conditions of Theorem 1 are satisfied. Hence, the fractional systems are stable for $J_1$. The Figures 1 and 2 show the related stability results for various values of '$\tilde{\alpha}$'.

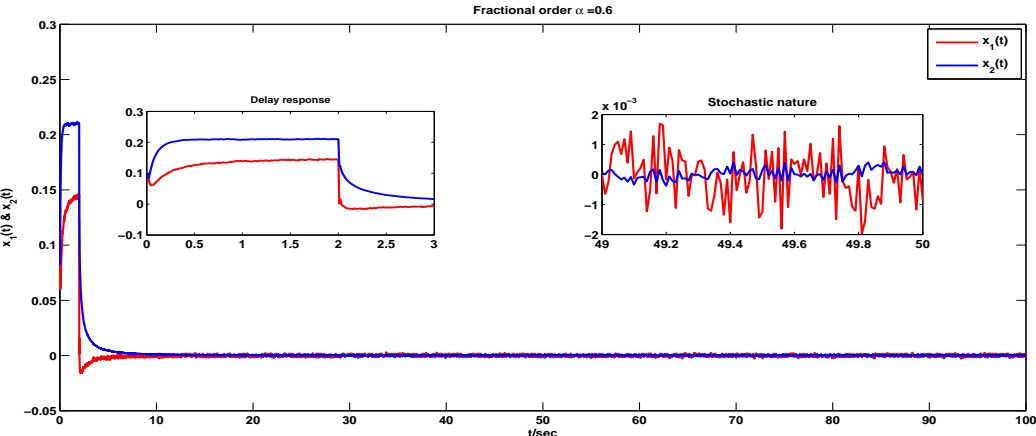

**Figure 1.** The systems (4)–(5) are stable at $\tilde{\alpha} = 0.6$.

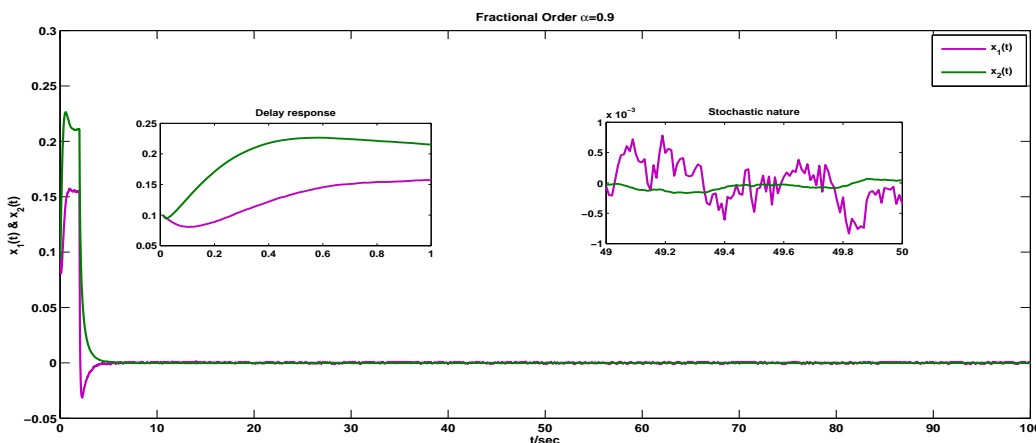

**Figure 2.** The systems (4)–(5) are stable at $\tilde{\alpha} = 0.9$.

Here, the delay response for the systems (4)–(5) is calculated for various values $\tilde{\alpha} = 0.6, 0.9$ and the delay occurred at t = 2. Further, the nonlinear functions $\tilde{f}_l, \tilde{\sigma}_l$ and $\tilde{g}_l$ are continuous and satisfy the assumption ($H_1$), and then using Theorem 1, the systems (4)–(5), they are stable on $[0, 100]$.

## 6. Conclusions and Future Research

In this paper, some useful and general conditions for exponential stability of NFSDS with fBm has been derived. The existence and uniqueness of fixed points, as well as the stability analysis of NFSDS, have been demonstrated. Finally, a numerical simulation was provided to demonstrate the theoretical findings. Based on the application of fractional-order stochastic financial modeling, the authors are interested in establishing the proposed model by considering the exponential stability of fractional stochastic delay systems with finance and stock price models and optimal control of stochastic insurance premium model in the near future.

**Author Contributions:** Conceptualization, O.S.H.; methodology, T.S.; software, T.A.; validation, O.S.H.; formal analysis, T.S.; investigation, T.S. and T.A.; resources, O.S.H. and T.S.; data curation, T.S.; writing—original draft preparation, T.S.; writing—review and editing, O.S.H.; visualization, T.S.; supervision, O.S.H.; project administration, T.S.; funding acquisition, O.S.H. and T.S. All authors have read and agreed to the published version of the manuscript.

**Funding:** This research received no specific grant from any funding agency in the public, commercial, or not-for-profit sectors.

**Institutional Review Board Statement:** Not applicable.

**Informed Consent Statement:** Not applicable.

**Data Availability Statement:** Not applicable.

**Acknowledgments:** The authors wish to thank the reviewers for their insightful comments which have greatly improved the paper.

**Conflicts of Interest:** The authors declare no conflict of interest. The funders had no role in the design of the study; in the collection, analyses, or interpretation of data; in the writing of the manuscript; or in the decision to publish the results.

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
