# Peer review of "Exponential Stability of Fractional Large-Scale Neutral Stochastic Delay Systems with Fractional Brownian Motion"

_jrfm, doi:10.3390/jrfm16050278_

Round 1

Reviewer 1 Report

Please find the attached report.

Author Response

Find the attached report

Reviewer 2 Report

This work concerns the exponential stability of fractional Large-Scale neutral stochastic delay systems. The results and their proofs are by now classical. They used a classical fixed point argument. The result lacks novelty and innovation, so I recommend rejecting its publication.

Author Response

See the attached report

Reviewer 3 Report

Please see the attached reviewer report.

Reviewer 4 Report

I have read this manuscript very carefully and found it interesting. In this manuscript, the authors discussed the exponential stability problem of fractional order Large-Scale neutral stochastic systems with delay. The existence and uniqueness of fixed points, as well as the stability analysis of NFSDS, have been demonstrated. Furthermore, authors provide numerical examples to support their theoretical results. So, I recommend this manuscript for publication.

Author Response

See the attached report

Round 2

Reviewer 3 Report

After reading the revised form of the manuscript, I am convinced that the authors have improved their previous version that making it now suitable for acceptance by the JRFM journal.